# Decreased Levels of Chaperones in Mucopolysaccharidoses and Their Elevation as a Putative Auxiliary Therapeutic Approach

**DOI:** 10.3390/pharmaceutics15020704

**Published:** 2023-02-20

**Authors:** Magdalena Żabińska, Lidia Gaffke, Patrycja Bielańska, Magdalena Podlacha, Estera Rintz, Zuzanna Cyske, Grzegorz Węgrzyn, Karolina Pierzynowska

**Affiliations:** Department of Molecular Biology, University of Gdansk, Wita Stwosza 59, 80-308 Gdansk, Poland

**Keywords:** mucopolysaccharidosis, chaperones, Hsp70, Hsp40, protein folding, ER stress, unfolded protein response

## Abstract

Mucopolysaccharidoses (MPS) are rare genetic disorders belonging to the lysosomal storage diseases. They are caused by mutations in genes encoding lysosomal enzymes responsible for degrading glycosaminoglycans (GAGs). As a result, GAGs accumulate in lysosomes, leading to impairment of cells, organs and, consequently, the entire body. Many of the therapies proposed thus far require the participation of chaperone proteins, regardless of whether they are therapies in common use (enzyme replacement therapy) or remain in the experimental phase (gene therapy, STOP-codon-readthrough therapy). Chaperones, which include heat shock proteins, are responsible for the correct folding of other proteins to the most energetically favorable conformation. Without their appropriate levels and activities, the correct folding of the lysosomal enzyme, whether supplied from outside or synthesized in the cell, would be impossible. However, the baseline level of nonspecific chaperone proteins in MPS has never been studied. Therefore, the purpose of this work was to determine the basal levels of nonspecific chaperone proteins of the Hsp family in MPS cells and to study the effect of normalizing GAG concentrations on these levels. Results of experiments with fibroblasts taken from patients with MPS types I, II, IIIA, IIIB, IIIC, IID, IVA, IVB, VI, VII, and IX, as well as from the brains of MPS I mice (*Idua*^−/−^), indicated significantly reduced levels of the two chaperones, Hsp70 and Hsp40. Interestingly, the reduction in GAG levels in the aforementioned cells did not lead to normalization of the levels of these chaperones but caused only a slight increase in the levels of Hsp40. An additional transcriptomic analysis of MPS cells indicated that the expression of other genes involved in protein folding processes and the cell response to endoplasmic reticulum stress, resulting from the appearance of abnormally folded proteins, was also modulated. To summarize, reduced levels of chaperones may be an additional cause of the low activity or inactivity of lysosomal enzymes in MPS. Moreover, this may point to causes of treatment failure where the correct structure of the enzyme supplied or synthesized in the cell is crucial to lower GAG levels.

## 1. Introduction

Mucopolysaccharidoses (MPS) are rare genetic diseases belonging to the lysosomal storage diseases (LSD). They appear with an average frequency of 1 in 25,000 live births. However, the frequency of their occurrence depends largely on the geographic region [1]. The direct cause of MPS are mutations in the genes encoding lysosomal enzymes responsible for the efficient degradation of glycosaminoglycans (GAG). These mutations cause an insufficient activity or a complete lack of activity of these enzymes, resulting in pathological accumulation of GAG in lysosomes, and thus, leading to a reduction in the efficiency of processes occurring within a single cell and the entire body [2].

Until recently, a total of 11 types and subtypes of this disease were distinguished (MPS I, II, IIIA, IIIB, IIIC, IIID, IVA, IVB, VI, VII, IX), depending on the defect of a particular enzyme and the type of stored GAG(s) (Table 1). Symptoms common to all types/subtypes of MPS include short stature, facial dysmorphism, chronic joint pain, organomegaly, or sensory problems [3]. Some types of MPS (severe cases of MPS I, MPS II, MPS VII, and all MPS III subtypes) may cause additional neurological disorders. These symptoms are common and can arise for two reasons: (i) direct damage to the central nervous system (CNS), caused by tissue destruction by GAG accumulation, or (ii) indirectly through the influence of somatic symptoms such as spinal cord compression or hydrocephalus, which in turn lead to brain damage resulting in aggression, sleep problems, speech difficulties, hearing loss, and personality changes [4,5]. Untreated patients with neurological symptoms lose the ability to learn new things over time and begin to regress in development, even losing previously acquired skills, which requires round-the-clock care [6]. The average life span of MPS patients varies between 10 and 20 years, depending on the type of the disease and severity of symptoms [1,3].

Recently, two new types of MPS (MPS X and MPS-PS) have been described (Table 1). MPS X is caused by mutations in the *ARSK* gene encoding arylsulfatase K, and, thus far, diagnosed patients have lived into their teens [7]. An interesting type of MPS seems to be MPS-PS syndrome, in which an elevated level of GAG is observed, but without a defect of any of the known lysosomal enzymes. The exact pathomechanism of this disease is unknown, despite the identification of mutations in the *VPS33A* gene, whose protein product is involved in endocytic and autophagic pathways. However, the efficiency of endocytosis and autophagy remains normal in this disease. MPS-PS patients most often die before the age of 2 years [8,9]. Unfortunately, cells from such patients are not available commercially.

Currently available treatment options for MPS include hematopoietic stem cell transplantation (HSCT) and enzyme replacement therapy (ERT), while gene therapy, pharmacological chaperone therapy, substrate reduction therapy (SRT), and therapy based on STOP-codon readthrough are also in the experimental phase. Unfortunately, these therapies do not lead to the cure of the disease, but only minimize its troublesome symptoms. Therefore, patients are doomed to emergency treatment, which requires constant involvement and cooperation of a group of various specialists (due to somatic complaints from various organs), as well as the patient’s carers [10,11,12].

Many of the therapies proposed thus far require the participation of chaperone proteins, regardless of whether they are therapies in common use (ERT) or remaining in the experimental phase (gene therapy, STOP-codon-readthrough therapy). Chaperones, which include heat shock proteins, are responsible for the correct folding of other proteins to the most energetically favorable conformation [13]. Without their appropriate level and activity, the correct folding of the lysosomal enzyme (introduced into the cell during ERT or synthesized in the cell after gene therapy or STOP-codon-readthrough therapy) might not be possible, and thus the activity of such an enzyme would be insufficient to eliminate the symptoms of the disease. Thus, if the level or activity of chaperones in MPS patients turned out to be too low, a lack of effectiveness of such therapy might be observed.

Our current knowledge about the baseline levels or activity of chaperones in MPS patients is scarce. The only report in this field indicated that impaired lysosomal activity caused massive perikaryal accumulation of insoluble α-synuclein and increased proteasomal degradation of cysteine string protein α (CSPα), two proteins thought to be chaperones, in MPS IIIA mouse models [14]. As a result, the availability of both α-synuclein and CSPα at the nerve terminals was significantly reduced, thereby inhibiting the assembly of the soluble NSF attachment receptor (SNARE) complex and the exchange of synaptic vesicles. It was indicated that restoring the correct levels of these proteins restored proteostasis and synaptic recycling, leading to the inhibition of neurodegeneration [14]. Indeed, it was indicated that proper protein turnover may be crucial to reduce neurodegeneration [15], while secondary changes in cellular processes contribute considerably to the pathomechanisms of MPS [16]. On the other hand, it must be considered that the two proteins mentioned above are chaperones specific to their target proteins. The baseline levels of the major, non-specific chaperone proteins, belonging to the Hsp family and involved in the folding of various enzymes, including lysosomal ones, have never been tested in the case of MPS.

Heat shock proteins (Hsps) are a large group of polypeptides not only involved in proper folding of other proteins, but also responsible for various stress responses [17,18,19]. Their levels increase significantly not only upon a quick increase in temperature, but also in response to other environmental stresses, such as appearance of toxic agents, enhanced oxidation, and others. Hsps can protect other proteins against denaturation and facilitate renaturation of already partially denatured molecules. However, when denaturation is impossible, Hsps expose such proteins (which are useless in cellular functions and potentially toxic to cells if forming large, insoluble aggregates) to proteolytic degradation. In fact, some Hsps are proteases themselves [17,18,19]. As suggested earlier, normal level of Hsps may be one of the key elements of effective therapies which are based on the delivery or synthesis of active enzymes. Therefore, the aim of this work was to investigate levels of the major non-specific chaperone proteins in the cells derived from patients suffering from MPS.

**Table 1 pharmaceutics-15-00704-t001:** Characteristics of MPS types/subtypes (based on ref. [16], with permission of the authors).

MPS Type/Subtype	Common Name	Stored GAG	Gene	Defective Enzyme
MPS I	Hurler, Scheie, Hurler–Scheie syndrome	HS, DS	*IDUA*	α-L-iduronidase
MPS II	Hunter syndrome	HS, DS	*IDS*	iduronate sulfatase
MPS IIIA	Sanfilippo type A syndrome	HS	*SGSH*	heparan sulfamidase
MPS IIIB	Sanfilippo type B syndrome	HS	*NAGLU*	N-acetylglucosaminidase
MPS IIIC	Sanfilippo type C syndrome	HS	*HGSNAT*	heparan-α-glucosaminide N-acetyltransferase
MPS IIID	Sanfilippo type D syndrome	HS	*GNS*	N-acetylglucosamine 6-sulfatase
MPS IVA	Morquio syndrome	KS, CS	*GALNS*	galactose-6-sulfate sulfatase
MPS IVB	Morquio syndrome	KS	*GLB1*	β-galactosidase
MPS VI	Maroteaux–Lamy syndrome	DS.	*ARSB*	N-acetylgalactosamine-4-sulfatase
MPS VII	Sly syndrome	HS, DS, CS	*GUSB*	β-glucuronidase
MPS IX	Natowicz syndrome	H	*HYAL1*	hyaluronidase
MPS X	-	DS	*ARSK*	arylsulfatase K
MPS-PS	MPS Plus syndrome	HS, DS	*VPS33A*	VPS33A

Abbreviations: HS—heparan sulfate, DS—dermatan sulfate, CS—chondroitic sulfate, H—hyaluronic acid.

## 2. Materials and Methods

### 2.1. Fibroblast Lines

In this study, fibroblast lines obtained from patients suffering from MPS types I, II, IIIA, IIIB, IIIC, IVA, IVB, VI, VII, and IX, and the HDFa line (a healthy control), were used. These lines were purchased from the Coriell Institute for Medical Research (Camden, NJ, USA). The exact characteristics of the cells are shown in Table 2. Cells were cultured under standard conditions (37 °C, grown in DMEM medium supplemented with 10% fetal bovine serum, 95% humidity, 5% CO_2_, and a standard antibiotic regimen).

### 2.2. Mouse Model of MPS Type I

A mouse MPS type I model (*Idua*^−/−^), purchased from The Jackson Laboratory (Bar Harbor, ME, USA; B6.129-Idua^tm1Clk^/J; #004068), was used in the study. The mice were kept as heterozygous inbred animals with constant access to water and food under standard conditions (temperature 22 °C, humidity 50–55%, day/night cycle 12 h/12 h). *Idua*^−/−^ (n = 6) and *Idua*^+/+^ (n = 6) mice at 6 months of age were sacrificed with Morbital (2 mL/kg), and livers and brains were collected. All experiments were carried out in accordance with the guidelines of the Council of the European Communities (2010/63/EU) and after approval by the Local Ethics Committee for Animal Experiments (Bydgoszcz, Poland).

### 2.3. Enzymes and Genistein

Aldurazyme (laronidase, recombinant human α-L-iduronidase) was from Sanofi-Genzyme (Cambridge, MA, USA) and used at a final concentration of 0.58 mg/mL. Elaprase (idursulfase, recombinant human 2-iduronate sulfatase) was from Shire Human Genetic Therapies (Cambridge, MA, USA) and used at a final concentration of 0.5 mg/mL. Genistein (5,7-dihydroxy-3-(4-hydroxyphenyl)-4H-1-benzopyran-4-one) was from Pharmaceutical Research Institute (Warsaw, Poland) and used at a final concentration of 50 µM. All these compounds were added to cell cultures, when indicated, for 24 h.

### 2.4. Western Blotting Analysis

In cellular experiments, fibroblasts (5 × 10^5^ cells) were passaged into 10 cm in diameter plates and left overnight. In some experimental variants, fibroblasts were incubated in the presence of ERT enzymes or genistein and DMSO (genistein control) for 24 h. Cells were then lysed in the lysis buffer (1% Triton X-100, 0.5 mM EDTA, 150 mM NaCl, 50 mM Tris, pH 7.5), supplemented with a mixture of protease and phosphatase inhibitors (Roche Applied Science, Penzberg, Germany). The protein lysate was obtained by microcentrifugation.

In animal tissue experiments, brains and livers were homogenized in the IG buffer (0.9% NaCl, 0.5% Triton X-100, 0.1% SDS, 1% sodium deoxycholate, 5 mM EDTA, 50 mM Tris-HCl, pH 7.5) [15] until obtaining a homogeneous mixture. Samples were transferred to new tubes and incubated for 2 h in the IG buffer in an ice bath with brief vortexing every 15 min. The samples were then centrifuged (13,000 rpm in a microcentrifuge) for 15 min. The obtained supernatants (protein lysates) were transferred to new tubes. 

The WES system (WES-Automated Western Blots with Simple Western; ProteinSimple, San Jose, CA, USA) for automatic Western blotting was used for protein separation. Proteins were separated using a 12–230 kDa separation module with 8 × 25 capillary cartridges (#SM-W004; ProteinSimple, San Jose, CA, USA). Hsp70 and Hsp40 were detected using the primary antibodies, rabbit anti-Hsp40 antibody (#4868, Cell Signaling Technology, Boston, MA, USA) and rat anti-Hsp70 antibody (#4872, Cell Signaling Technology, Boston, MA, USA), respectively, both diluted 1:50. Secondary anti-rabbit antibodies from anti-rabbit detection module (No. DM-001, ProteinSimple, San Jose, CA, USA) or secondary anti-rat antibodies (#31470, Invitrogen, Waltham, MA, USA) were used, respectively, for signal detection. Protein levels were determined by WES using the total protein chemiluminescence detection module (#DM-TP01, ProteinSimple, San Jose, CA, USA) as a loading control.

### 2.5. Fluorescent Microscopy

Cells (5 × 10^4^) were cultured on coverslips overnight after they were allowed to attach to the glass surface. After a 24-hour incubation in the presence of ERT enzymes, genistein, or DMSO (genistein control), fixation of fibroblasts with 2% paraformaldehyde in phosphate buffered saline (PBS) was performed, and followed by washing with 0.1% Triton X-100 in PBS. The BSA (5%) buffer was then used for the blocking reaction for 1 h, followed by overnight incubation with the primary antibody (anti-Hsp40 (diluted 1:400) or anti-Hsp70 (diluted 1:100), purchased from Cell Signaling Technology, Boston, MA, USA (#4868 and #4872, respectively). Cells were washed 3 times with PBS and incubated with secondary antibodies (Alexa fluor 488 (diluted 1:500), Thermo Fisher Scientific, Waltham, MA, USA; A32731) for 1 h. Another washing procedure was performed 5 times with PBS, and then coverslips were mounted on slides (using a mounting medium). Then, they were analyzed under a fluorescence microscope (Leica, Wetzlar, Germany).

### 2.6. GAG Level Measurement

Cellular GAG levels were measured in the obtained cell lysates using the Glycosaminoglycan Assay Blyscan™ kit (Biocolor Life Science Assays, Carrickfergus, UK). The GAG levels were calculated according to the manufacturer’s instructions.

### 2.7. Transcriptomic Analyses

Transcriptomic analyses were performed according to a previously described procedure [16]. Briefly, fibroblasts (5 × 10^5^ cells) were passaged in plates (10 cm diameter) overnight. Cell lysis was performed in a buffer containing guanidine isothiocyanate and β-mercaptoethanol. The cells were then homogenized using a QIAshredder column according to the manufacturer’s instructions (Qiagen, Hilden, Germany). Ribonucleic acids were extracted using the RNeasy Mini kit (Qiagen, Hilden, Germany) and DNA contaminants were removed using Turbo DNase (Life Technologies, Carlsbad, CA, USA). The quality of the obtained RNA samples was tested with the Nano Chips RNA kit (Agilent Technologies, Santa Clara, CA, USA) using the Agilent 2100 Bioanalyzer System. Isolation of nucleic acids from each cell line was performed 4 times from independent cultures.

To obtain the mRNA libraries, the Illumina TruSeq Stranded mRNA Library Prep Kit was employed. Following the reaction with reverse transcriptase, sequencing of cDNA libraries was conducted with HiSeq4000 (Illumina, San Diego, CA, USA) (PE150; paired end 150 bp). About 40 million raw readings were obtained (each sample of about 12 Gb of raw data). FastQC (version v0.11.7) was employed for the quality assessment. To map the raw reads (in comparison to the human reference GRCh38 genome), the Ensembl database and Hisat2 v. 2.1.0 software were used. Cuffquant and Cuffmerge (version 2.2.1) were employed to calculate transcript levels with the use of the GTF file Homo_sapiens.GRCh38.94.gtf from the Ensembl database (https://www.ensembl.org/index.html; 5 December 2021). Cuffmerge was launched with the parameter library-norm-method classic-fpkm, normalizing expression values using the FPKM algorithm. The raw data are available in the NCBI Sequence Read Archive (SRA) at the accession no. PRJNA562649.

### 2.8. Statistical Analyses

R v3.4.3 software was used for statistical analyses. In transcriptomic studies, statistical significance between the two normally continuous groups, with log_2_(1 + x) values, was assessed by one-way ANOVA and post hoc Student’s *t*-test with Bonferroni correction. The Benjamini–Hochberg method was used to calculate the false discovery rate (FDR). The Ensembl gene database (BioMart interface; https://www.ensembl.org/info/data/biomart/index.html; 21 December 2022) was used to classify the transcripts. In other experiments, two-way ANOVA and post hoc Tukey’s test was employed.

## 3. Results

Maintaining the correct levels of chaperone proteins is crucial for cell proteostasis. Incorrect levels or activity of molecular chaperones may result in the accumulation of misfolded proteins, which in turn leads to endoplasmic reticulum (ER) stress, followed by an unfolded protein response (UPR). In addition, in the case of MPS, abnormalities related to the structure and function of the chaperones may result in a lack of effectiveness of therapies involving the delivery of missing enzymes during ERT or the synthesis of enzymes in the cells during gene therapy or STOP-codon readthrough, because these enzymes without chaperone proteins may not be able to achieve the appropriate conformations, and thus to fulfill their functions. Therefore, we aimed to estimate the levels of major chaperones in MPS cells.

### 3.1. The Level of Major Chaperone Proteins from the Hsp Family in MPS Cells

Studies conducted on fibroblasts taken from patients with MPS type I and II indicated a drastic reduction in the levels of major chaperone proteins from the Hsp family (Hsp70 in MPS I and II at *p* < 0.001; Hsp40 in MPS I *p* < 0.001 and MPS II at *p* = 0.007), reaching about 15% of those in control cells, as indicated by Western blotting immunodetection analyses (Figure 1A,B and Figure 2A,B) and by fluorescence microscopy (Figure 1C,D and Figure 2C,D). As expected, GAG levels were significantly increased in MPS I (*p* < 0.001) and MPS II (*p* = 0.002) cells relative to the control fibroblasts (HDFa) (Figure 3). Animal model studies confirmed the results obtained with cell lines when the MPS I mouse brain was investigated (i.e., decreased Hsp70 (*p* = 0.038) and Hsp40 (*p* = 0.007) levels). In contrast to the brain, liver was characterized by the opposite effect, i.e., an increase in the level of Hsp70 (*p* = 0.016) and no significant changes in the level of Hsp40 (*p* = 0.189) in *Idua*^−/−^ mice as compared to the control mice (Figure 4).

Next, we investigated whether reduction in GAGs in MPS cells utilizing Aldurazyme (MPS I), Elaprase (MPS II), or genistein (an SRT-based treatment) can correct observed decreased levels of the tested Hsp proteins. We found that reduction in GAG levels (Figure 3) by the aforementioned compounds (Aldurazyme and genistein in MPS I at *p* < 0.001; Elaprase in MPS II at *p* = 0.035, genistein in MPS II at *p* = 0.046) resulted in only a partial increase in the levels of the Hsp40 protein (Aldurazyme in MPS I at *p* = 0.007, genistein in MPS I at *p* = 0.003, Elaprase in MPS II at *p* = 0.049, genistein in MPS II at *p* = 0.047) (Figure 2), without affecting the levels of Hsp70 (Aldurazyme in MPS I at *p*=0.064, genistein in MPS I at *p* = 0.078, Elaprase in MPS II at *p* = 0.063, genistein in MPS II at *p* = 0.128) (Figure 1).

### 3.2. Transcriptomic Analysis of Changes in the Processes Involving Chaperones in MPS Cells

Hsp70 and Hsp40 are major chaperone proteins, but not the only ones. To identify changes in the expression of genes encoding various molecular chaperones and/or proteins related to the functions or structures of chaperones, we performed a transcriptomic analysis of cells derived from patients with MPS types I, II, IIIA, IIIB, IIIC, IIID, IVA, IVB, VI, VII, and IX. The results of these analyses, performed with the use of the Gene Ontology (GO) database and assessing GO terms, indicated significant changes in the expression of genes involved in processes involving chaperone proteins, such as protein folding (GO:0006457), response to endoplasmic reticulum stress (GO:0034976), and response to unfolded protein (GO:0006986) (Figure 5). Further analysis of transcripts whose expression was changed in at least five types of MPS allowed us to identify genes encoding chaperones (*HSPB6*, *CLU*, *DNAJC3*), ER proteins interacting with chaperones and/or influencing their activity (*PRKCSH*, *TMX1*), or proteins that are components of the complex involved in the degradation of misfolded proteins (*UFD1*) (Table 3). 

We also found that the levels of some transcripts were particularly strongly altered in MPS cells. Considering the logarithm value of the fold change (log_2_FC), we observed that the expression of over 10 genes was up-regulated or down-regulated (at *p* < 0.1) by more than four times (log_2_FC > 2 or log_2_FC < −2) (Figure 6). The list of such genes includes those encoding chaperone proteins (*HSPB6*, *CLU*, *HSPB8, HSPB7*), ER proteins interacting with chaperone proteins and/or affecting their activity (e.g., *KDELR3*), but also chemokines (e.g., *CXCL8*), lamins (e.g., *LMNA*), caveola plasma membranes (e.g., *CAV1*), extracellular matrix proteins (e.g., *COMP*), and proteins involved in chaperone-mediated autophagy (e.g., *BBC3*) (Table 4).

In addition, the transcriptomic analyses were visualized using the KEGG pathway database (protein processing in endoplasmic reticulum pathways) (Figure 7). This visualization indicates changes in specific processes, either up- or down-regulated in MPS.

## 4. Discussion

Proper folding of proteins into their correct three-dimensional and functional forms is essential for the maintenance of proteostasis and, consequently, for the proper functioning of cells, tissues, and organs. Chaperones, involved in the stabilization, folding, and unfolding of client proteins, have therefore long been at the center of researchers’ interest in issues of protein balance and the consequences of its disruption. The Hsp family of proteins, constituting the most common among chaperones, is a highly conserved group of polypeptides that (i) assist other proteins during folding, (ii) cause their re-folding after partial denaturation, (iii) facilitate their movement to target locations of actions, (iv) expose them to proteasomal degradation, and (v) take part in chaperone-mediated autophagy [17,18]. Thus, it is not surprising that the roles played by chaperone proteins are irreplaceable, and any disruption of their levels or activities contribute to many serious human diseases [19,20,21,22,23,24,25]. Therefore, it is not surprising that modulation of chaperone protein levels or the use of pharmacological chaperones, small molecules that mimic chaperone proteins in their effects on folding of other proteins, has been proposed as the main or adjunct therapy for various disorders, including LSDs [26,27].

Since LSDs are caused by a lack of lysosomal enzyme activity that is a consequence of a genetic mutation, the administration of a pharmacological chaperone that would be involved in the proper retraction of such enzymes seems to make the most sense. Pharmacological chaperones have a very specific action with respect to their client proteins, and studies on their effectiveness are very promising but still in the clinical or preclinical research phase (e.g., migalastat in Fabry disease, ambroxol in Gaucher disease, miglustat in Pompe disease, N-octyl-4-epi-β-valienamine (NOEV) in gangliosidosis, and others) [27,28].

In the case of MPS, a number of compounds have already been proposed for chaperone therapy, including Δ-unsaturated 2-sulfouronic acid-N-sulfoglucosamine, which effectively increased IDS activity in MPS II, but without affecting GAG levels [29]. MPS IIIC cells were also characterized by increased heparan acetyl-CoA:alpha-glucosaminide N-acetyltransferase (HGSNAT) activity under the influence of glucosamine [30]. An increase in the activity of human recombinant GALNS (hrGALNS) (for MPS IV) produced in bacteria (*Escherichia coli*), yeasts (*Pichia pastoris*), and mammalian cells (HEK293) was also observed after incubation of cells in the presence of ezetimibe and pranlukast [31]. However, it should be borne in mind that this therapy involves the administration of a specific pharmacological chaperone against a specific lysosomal enzyme in a single type of MPS, which is assumed to properly fold and prevent degradation of the enzyme, in which even a small increase in activity can alleviate some disease symptoms. However, it is necessary to underline that pharmacological chaperones are small molecules, while, to our knowledge, the baseline levels of molecular chaperones (being proteins) have not been studied in MPS to date.

It is important to maintain adequate levels of chaperones in MPS cells, and thus to facilitate the process of protein folding, as shown by a few research reports. It has been pointed out that the appearance of mutations prevents the enzyme from achieving its correct structure and proper glycosylation which results in mistargeting the enzyme. Namely, instead of its transport to the lysosome, it retains in ER, and then it is directed to ER-associated degradation [30,32]. Silencing the expression of genes encoding ERAD components [32,33], the use of a competitive inhibitor of the enzyme [30], and increasing the levels of chaperone proteins were all be able to increase the level as well as the activity of the enzyme [33]. 

It was also proposed that the enzyme folding process plays an important role in its stability and targeting to lysosomes. An increased residual NAGLU activity in skin fibroblasts of MPS IIIB patients cultured at 30 °C, rather than at 37 °C, was demonstrated, suggesting different efficiencies of chaperone proteins at both temperatures [34]. The authors of that report hypothesized that higher levels of NAGLU in MPS IIIB cells at 30 °C than at 37 °C might arise from the presence of more abundant molecular chaperones at the former temperature [34]. Such a proposal appears quite controversial in the light of commonly accepted evidence that expression of heat shock genes is stimulated at higher temperatures [35]. Therefore, we propose an alternative hypothesis that, at higher temperatures (such as 37 °C) there is an elevated levels of Hsps relative to lower temperatures (such as 30 °C). However, according to known activities of molecular chaperones, they act firstly to re-fold mutant proteins and to fold them properly, but if this is not possible and they fail, then improperly folded or damaged polypeptides are directed into the proteolysis [13]. Hence, increased levels of Hsps at 37 °C might cause enhanced proteolysis of improperly folded NAGLU variants, lowering their residual enzymatic activities, while at 30 °C the mutant variants could be more stable due to less-efficient chaperone-facilitated proteolysis. 

The link between the Hsps and lysosomal enzyme activity in MPS was demonstrated by showing an increase in Hsp70 levels and increased α-L-iduronidase (IDUA) activity, leading to a decrease in GAG levels after gentamicin treatment (STOP-codon-readthrough therapy) [36]. Moreover, after dithyldiazem treatment, increased expression of genes coding for chaperones, such as BiP and Hsp40, and increased ability of ER to fold client proteins by regulating calcium levels, could be observed in models of different lysosomal diseases [37]. The examples presented above demonstrated that chaperone proteins significantly affect not only the activity of the lysosomal enzyme but also its transport to its target location of action, the lysosome.

Both the levels and activities of chaperones are extremely important in the context of current therapies. The delivery of the recombinant enzyme to the patient’s body (ERT) and the synthesis of enzymes after gene therapy or STOP-codon-readthrough therapy require the presence of chaperones for the lysosomal enzyme molecule to achieve a functional structure. Based on the known functions of Hsps [17,18,19], one might assume that the use of the above-mentioned therapies at too-low levels of chaperones in patients results in the presence of the enzyme but its improper structure, due to an inefficient protein quality control system. This could provide a partial explanation for the failures of these therapies to correct all the disease symptoms. Therefore, the aim of the present study was to determine the levels of chaperone proteins in the cells of MPS patients, and to test how these levels are affected by the two GAG-lowering therapies.

The results presented in this work indicated significantly decreased levels of two non-specific, major chaperone proteins, Hsp40 and Hsp70, in cells derived from MPS patients and in the brains of mice representing a model of MPS type I (*Idua*^−/−^) (Figure 1 and Figure 2). Even more interestingly, none of the treatments used to reduce GAG levels restored the Hsp proteins’ levels to those measured in control cells, despite some increases being observed for Hsp40 after incubating the cells in the presence of recombinant enzymes (Aldurazyme for MPS I and Elaprase for MPS II) and genistein (used in substrate synthesis reduction therapy, SRT). Using transcriptomic analysis, the present study led to the identification of genes whose products are associated with protein folding processes and which showed altered expressions in MPS cells. These include not only heat shock proteins themselves, but also other proteins involved in the folding of client proteins. One of them is the *CLU* gene, encoding clusterin, which we have already described in the context of MPS pathogenesis [38] (Table 3 and Table 4). High expression levels of this gene were also confirmed by other research teams, proposing it as a diagnostic marker for mucopolysaccharidosis-related arterial disease in MPS I [39] or bone and cartilage metabolic disorders in MPS IV [40].

Studies on the transcriptome, associated with the unfolded protein response, were also conducted previously with cells from patients with different types/subtypes of MPS [41]. No activation of the unfolded protein response (UPR) was found due to the lack of changes in the expression of essential genes associated with this response (Xbp, BiP, Chop, Edem1, Edem2, Edem3) in the case of lesioned cells [41]. Our transcriptomic analysis did indicate an increased expression of the *Edem1* gene in MPS IIIB (log_2_FC = 0.54) but not in lines of other types/subtypes. Activation of UPR is thought to (i) restore normal cell functions by stopping protein translation, (ii) increase the production of molecular chaperone proteins involved in the protein folding, and (iii) degrade misfolded proteins. To the extent that all these processes are dependent on each other, one could speculate that too-low levels of certain non-specific chaperone proteins (such as Hsp40 and Hsp70) insufficiently activate UPR which, in turn, could affect protein deposition and autophagy–lysosomal pathway impairment [42].

Finally, it might be surprising that, although amounts of both tested molecular chaperones were evidently decreased in MPS fibroblasts (Figure 1 and Figure 2), changes in levels of Hsp70 and Hsp40 proteins were different in the brains and livers of MPS I mice (Figure 4). However, different expressions of gene coding for Hsps in different organs of experimental animals have been reported previously, especially under stress conditions [43]. Since GAG storage causes permanent stress conditions in MPS cells and organs, it is likely that the results obtained in this study reflect the processes reported by others [43] which are based on variability of the efficiencies of stress responses in different organs due to their differential physiology and sensitivity to various agents and factors. 

## 5. Conclusions

In conclusion, reduced levels of major chaperone proteins Hsp70 and Hsp40 may be an additional cause of low activity or inactivity of the deficient enzymes in MPS diseases. Moreover, this may point to causes of treatment failure where the correct structure of the enzyme supplied to or synthesized in the cells is crucial to lowering GAG levels. Thus, we propose to consider whether (together with ERT, gene therapy, or STOP-codon-readthrough therapy) a solution aimed at increasing the level of chaperone proteins should be used. Such a solution is used, among others, for cancer cells where administration of Hsp90 inhibitors (geldanamycin, 17-N-allylamino-17-demethoxygeldanamycin (17AAG), or 17-dimethylaminoethylamino-17-demethoxygeldanamycin (17DMAG)) leads to an increase in Hsp70 levels [44], or, as mentioned, for brain injury, where administration of geranylgeranylacetone (GGA) also led to an increase in Hsp70 levels [20]. It is also possible to use compounds that affect the increase in efficiency of expression of genes encoding heat shock proteins in combination with other applied therapies for MPS [45]. We believe that solutions of this type could favorably affect the activity of lysosomal enzymes in MPS, which is crucial to improve patients’ conditions, as even a small increase in the activity of the deficient enzyme can lead to a considerable alleviation of symptoms of the disease.

## Figures and Tables

**Figure 1 pharmaceutics-15-00704-f001:**
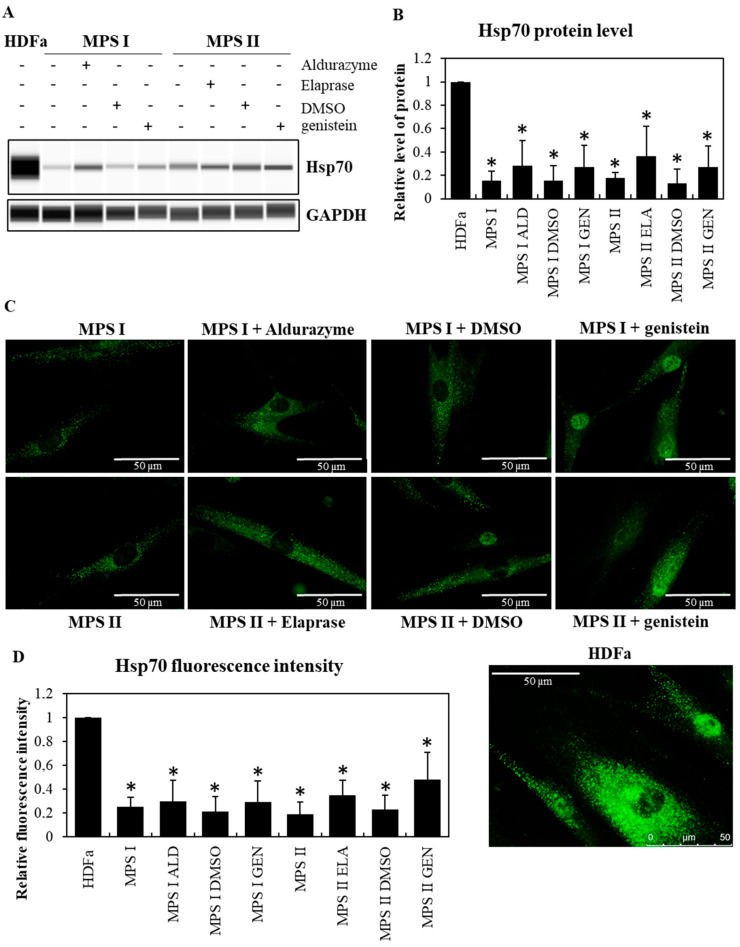
Decreased levels of Hsp70 in MPS cells and effects of ERT and SRT. Western blotting (**A**,**B**) and immunofluorescence (**C**,**D**) analyses of the abundance of Hsp70 in MPS I and MPS II fibroblasts, relative to HDFa control cells, either non-treated (minuses) or treated (pluses) with Aldurazyme (ALD, recombinant α-L-iduronidase) at 0.58 mg/mL, Elaprase (ELA, recombinant 2-iduronate sulfatase) at 0.5 mg/mL, DMSO (solvent for genistein) at 0.05%, or genistein (GEN) at 50 μM for 24 h. Panel (**A**) shows a representative Western blot (with GAPDH as loading control) while panel (**B**) demonstrates quantitative analyses (based on densitometry) from 3 independent experiments, with error bars representing SD. Panel (**C**) shows representative fluorescent microscopic pictures (with scale bars indicating 50 μm), while panel (**D**) demonstrates quantitative analyses (relative fluorescence intensity) from 100 randomly chosen cells of each variant of the experiment. In panels (**B**) and (**D**), asterisks indicate statistically significant differences (at *p* < 0.05 in two-way ANOVA and post hoc Tukey’s test) relative to HDFa cells. No statistically significant differences were detected between non-treated and treated MPS I or MPS II fibroblasts.

**Figure 2 pharmaceutics-15-00704-f002:**
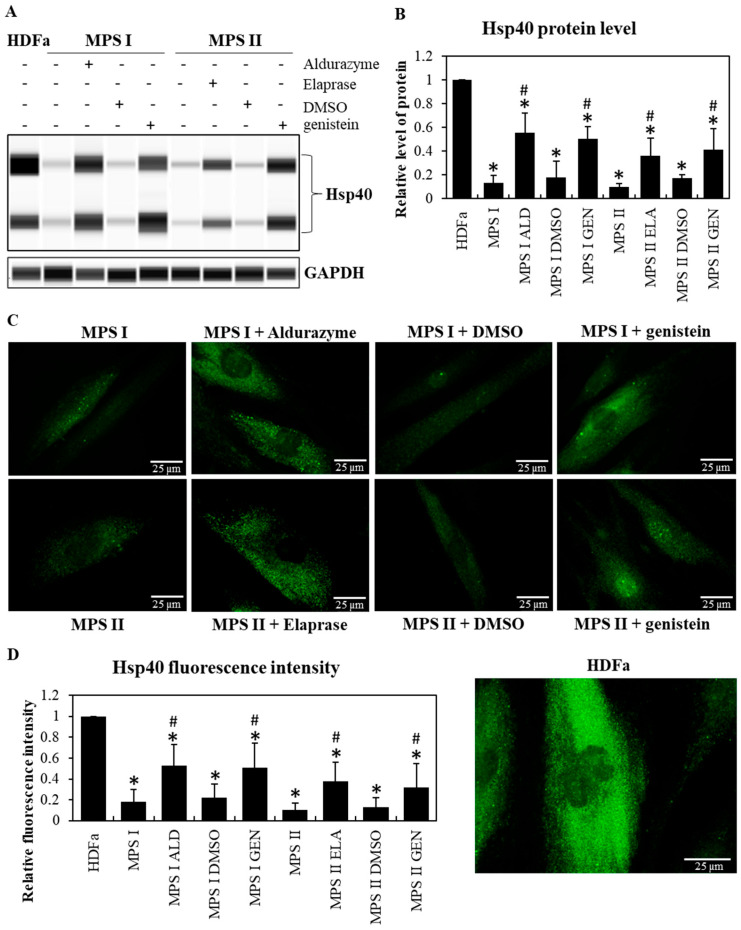
Decreased levels of Hsp40 in MPS cells and effects of ERT and SRT. Western blotting (**A**,**B**) and immunofluorescence (**C**,**D**) analyses of the abundance of Hsp40 in MPS I and MPS II fibroblasts, relative to HDFa control cells, either non-treated (minuses) or treated (pluses) with Aldurazyme (ALD, recombinant α-L-iduronidase) at 0.58 mg/mL, Elaprase (ELA, recombinant 2-iduronate sulfatase) at 0.5 mg/mL, DMSO (solvent for genistein) at 0.05%, or genistein (GEN) at 50 μM for 24 h. Panel (**A**) shows a representative Western-blot (with GAPDH as loading control) while panel (**B**) demonstrates quantitative analyses (based on densitometry) from 3 independent experiments with error bars representing SD. Panel (**C**) shows representative fluorescent microscopic pictures (with scale bars indicating 25 μm), while panel (**D**) demonstrates quantitative analyses (relative fluorescence intensity) from 100 randomly chosen cells of each variant of the experiment. In panels (**B**) and (**D**), asterisks indicate statistically significant differences (at *p* < 0.05) relative to HDFa cells, and hashtags indicate statistically significant differences (at *p* < 0.05 in two-way ANOVA and post hoc Tukey’s test) relative to non-treated MPS I or MPS II fibroblasts.

**Figure 3 pharmaceutics-15-00704-f003:**
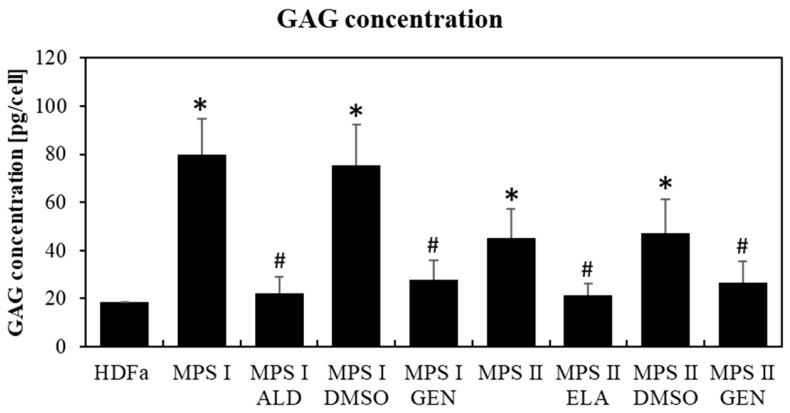
Elevated GAG levels in MPS cells and storage reduction by ERT or SRT. GAG levels in MPS I and MPS II fibroblasts, relative to HDFa control cells, either non-treated or treated with Aldurazyme (ALD, recombinant α-L-iduronidase) at 0.58 mg/mL, Elaprase (ELA, recombinant 2-iduronate sulfatase) at 0.5 mg/mL, DMSO (solvent for genistein) at 0.05%, or genistein (GEN) at 50 μM for 24 h. Asterisks indicate statistically significant differences (at *p* < 0.05 in one-way ANOVA) relative to HDFa cells, and hashtags indicate statistically significant differences (at *p* < 0.05 in two-way ANOVA and post hoc Tukey’s test) relative to non-treated MPS I or MPS II fibroblasts.

**Figure 4 pharmaceutics-15-00704-f004:**
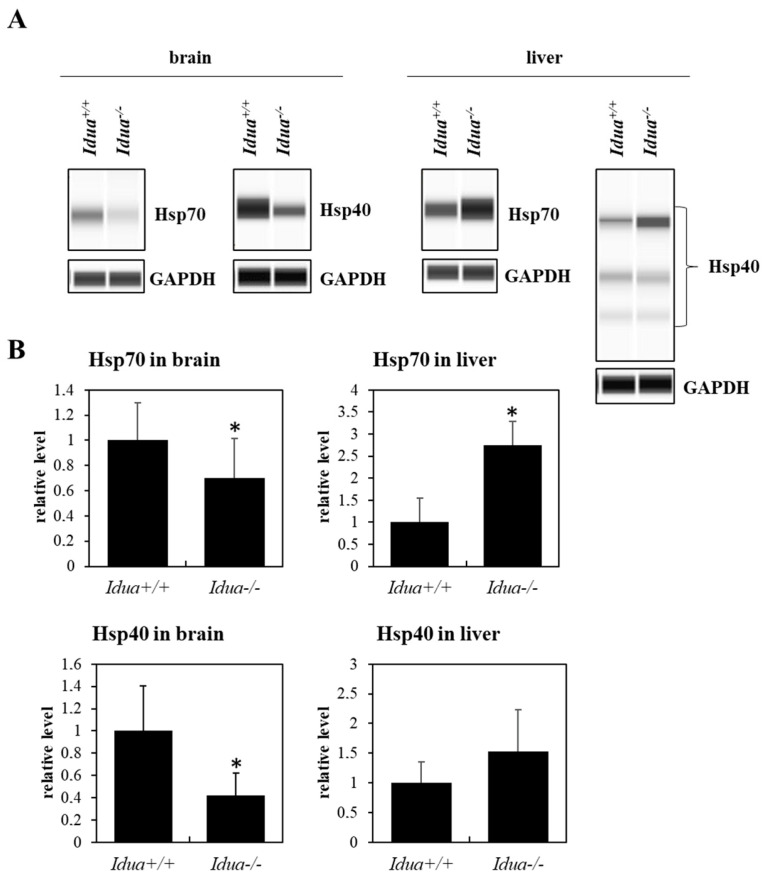
Hsp70 and Hsp40 levels in brain and liver tissues of MPS I mice. Western blotting analyses (with GAPDH as a loading control) of the abundance of Hsp70 and Hsp40 in the brains and livers of MPS I (*Idua*^−/−^) and control (*Idua*^+/+^) mice. Representative blots are shown (**A**), and quantitative analyses (based on densitometry) from 6 animals in each group, with error bars representing SD, are demonstrated (**B**). Asterisks indicate statistically significant differences (at *p* < 0.05 in two-way ANOVA and post hoc Tukey’s test) relative to control (*Idua*^+/+^) mice.

**Figure 5 pharmaceutics-15-00704-f005:**
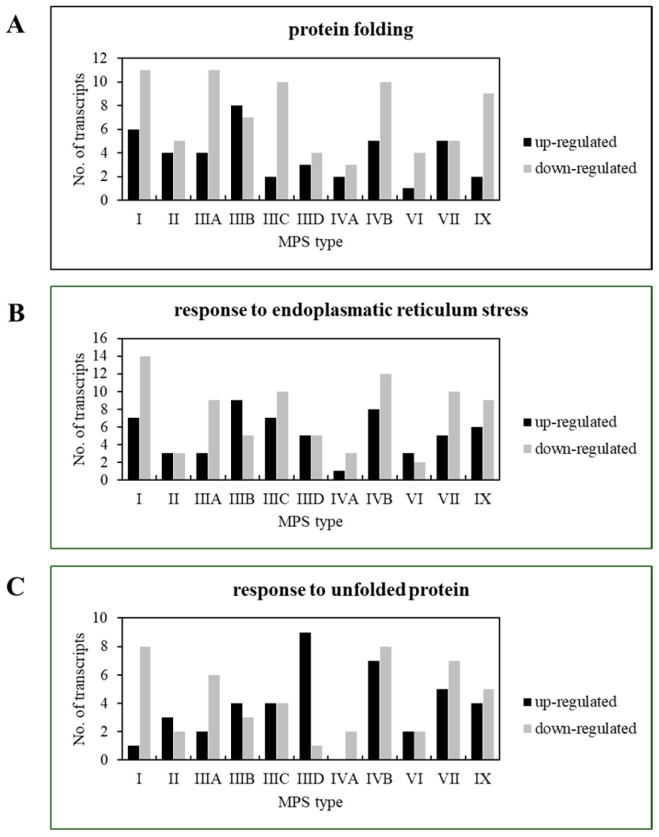
Changes in expression of genes coding for proteins involved in protein folding (**A**), response to endoplasmic reticulum stress (**B**), and response to unfolded protein (**C**) in MPS cells. Transcriptomic analyses indicating up- and down-regulation of genes represented in indicated Gene Ontology pathways (terms) in tested MPS fibroblasts relative to HDFa control cells. The number of genes of each class with significantly affected expression in MPS cells is shown in every panel.

**Figure 6 pharmaceutics-15-00704-f006:**
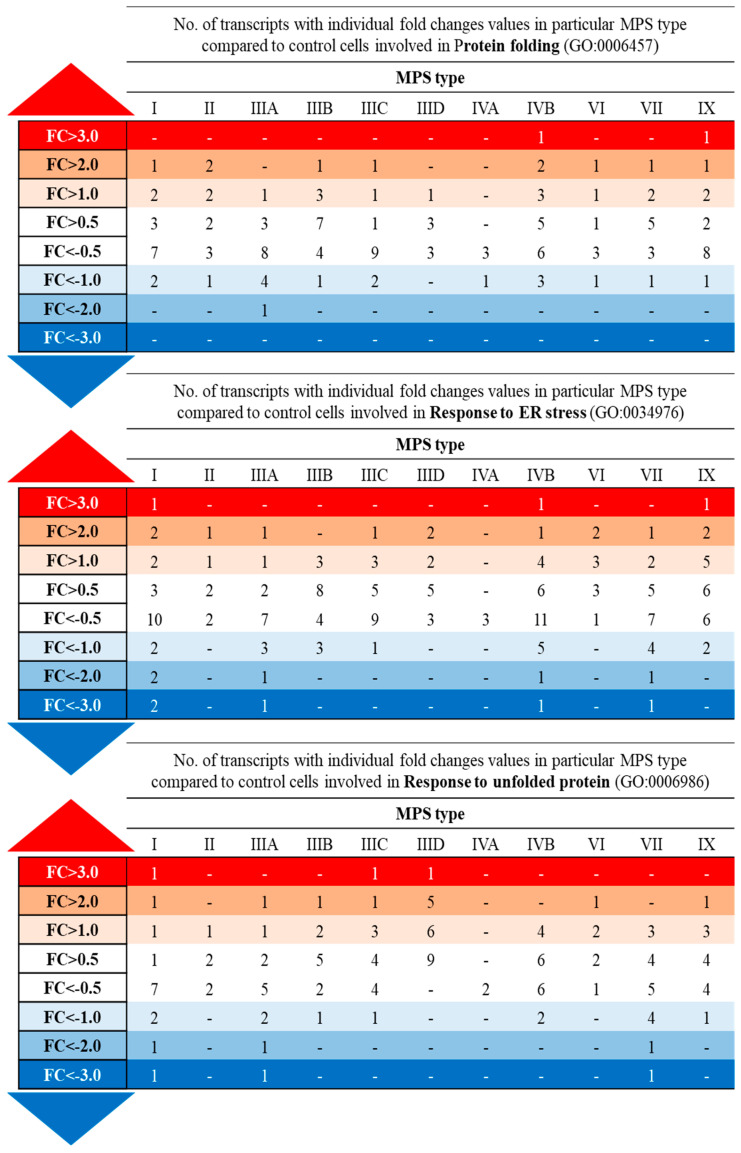
Number of transcripts included into indicated Gene Ontology terms with altered expression (*p* < 0.1 in one-way ANOVA and post hoc Student’s *t*-test with Bonferroni correction) depending on the level of fold-change (FC) value in different types of MPS relative to HDFa control cells.

**Figure 7 pharmaceutics-15-00704-f007:**
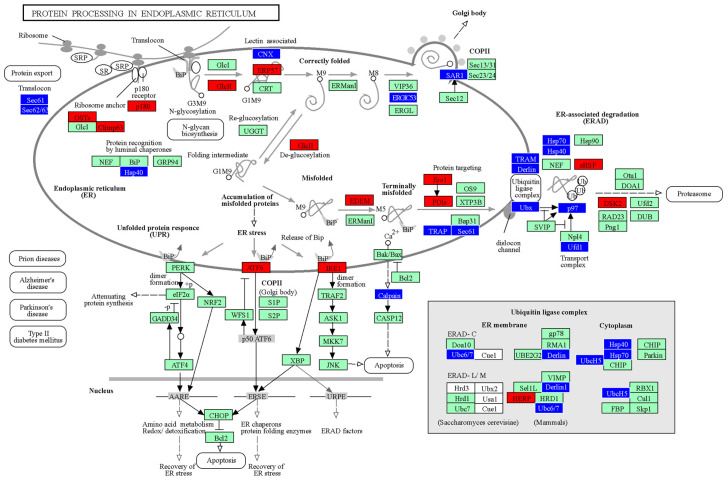
Changes in protein processing in MPS cells. The KEGG pathway presenting the ‘protein processing in endoplasmic reticulum’ process, imaged from transcriptomic data derived from MPS cells. Individual proteins or groups of proteins were colored if a change in the expression of an indicated gene was observed in at least one type/subtype of MPS. Down-regulated and up-regulated genes are marked in blue and red, respectively. Non-marked values indicate results in which no statistically significant differences between MPS and HDFa were determined.

**Table 2 pharmaceutics-15-00704-t002:** Characteristics of MPS patient-derived fibroblast lines (based on ref. [16], with permission of the authors).

MPS Type/Subtype	Sex	Age *	Mutated Gene and Mutations	Cat. No. ^#^
MPS I	Female	1	*IDUA*, p.Trp402X/p.Trp402X	GM00798
MPS II	Male	3	*IDS*, p.His70ProfsX29/-	GM13203
MPS IIIA	Female	3	*SGSH*, p.Glu447Lys/p.Arg245His	GM00879
MPS IIIB	Male	7	*NAGLU*, p.Arg626Ter/p.Arg626Ter	GM00156
MPS IIIC	Male	8	*HGSNAT*, p.Gly262Arg/pArg509Asp	GM05157
MPS IIID	Male	7	*GNS*, p.Arg355Ter/p.Arg355Ter	GM05093
MPS IVA	Female	7	*GALNS*, p.Arg386Cys/p.Phe285Ter	GM00593
MPS IVB	Female	4	*GLB1*, p.Trp273Leu/p.Trp509Cys	GM03251
MPS VI	Female	3	*ARSB*, ND	GM03722
MPS VII	Male	3	*GUSB*, p.Trp627Cys/p.Arg356X	GM00121
MPS IX	Female	14	*HYAL1*, ND	GM17494

Abbreviations: * age at the time of sample collection; # Coriell Institute catalogue number; ND, not determined (diagnosis made based on elevated GAG levels and drastically decreased enzyme activity).

**Table 3 pharmaceutics-15-00704-t003:** Genes included into ‘protein folding’ (GO:0007165), ‘response to endoplasmic reticulum stress’ (GO:0034976), and ‘response to unfolded protein’ (GO:0006986) terms of Gene Ontology with expression significantly changed (at FDR < 0.1; *p* < 0.1 in one-way ANOVA and post hoc Student’s *t*-test with Bonferroni correction) in most (5 or more) MPS types relative to control cells (HDFa). Down-regulated and up-regulated genes are marked in blue and red, respectively. Non-marked values indicate results in which no statistically significant differences between MPS and HDFa were determined.

Transcript	log_2_FC of Selected Transcripts’ Levels in Particular MPS Type vs. HDFa Line										
	I	II	IIIA	IIIB	IIIC	IIID	IVA	IVB	VI	VII	IX
Protein folding (GO:0006457)											
*HSPB6*	1.90	2.03	0.14	2.29	2.27	0.38	1.42	2.37	1.47	1.55	1.57
*PDIA3*	0.46	0.39	0.31	0.50	0.30	0.38	0.37	0.36	0.20	0.32	−0.13
*CLU*	2.69	2.98	1.59	1.75	2.95	1.14	1.73	3.46	2.58	2.08	3.44
*PRKCSH*	0.42	0.45	0.52	0.50	0.38	0.33	0.50	0.51	0.22	0.39	0.09
*DNAJC3*	−0.58	−0.58	−0.56	−0.60	−0.86	−0.37	−0.58	−0.71	−0.55	−0.15	−0.63
Response to endoplasmic reticulum stress (GO:0034976)											
*PDIA3*	0.46	0.39	0.31	0.50	0.30	0.38	0.37	0.36	0.20	0.32	−0.13
*TRIM25*	0.60	0.39	0.29	0.88	0.51	0.69	0.27	1.00	0.29	0.79	0.94
*CLU*	2.69	2.97	1.59	1.75	2.95	1.14	1.73	3.46	2.58	2.08	3.44
*TMX1*	−0.75	−0.47	−0.86	−0.94	−0.62	−0.84	−0.54	−0.51	−0.51	−1.06	−1.07
*DNAJC3*	−0.58	−0.58	−0.56	−0.60	−0.86	−0.37	−0.58	−0.71	−0.55	−0.15	−0.63
*UFD1*	−0.50	−0.38	−0.61	−0.44	−0.59	−0.62	−0.66	−0.51	−0.40	−0.55	−0.78
Response to unfolded protein (GO:0006986)											
*DNAJC3*	−0.58	−0.58	−0.56	−0.60	−0.86	−0.37	−0.58	−0.71	−0.55	−0.15	−0.63

**Table 4 pharmaceutics-15-00704-t004:** Genes included into ‘protein folding’ (GO:0007165), ‘response to endoplasmic reticulum stress’ (GO:0034976), and ‘response to unfolded protein’ (GO:0006986) terms of Gene Ontology with expression significantly changed (at FDR < 0.1; *p* < 0.1 in one-way ANOVA and post hoc Student’s *t*-test with Bonferroni correction) and with particularly high values of the fold change (log_2_FC > 2 or <−2) in specific MPS types relative to control cells (HDFa). Down-regulated and up-regulated genes are marked in blue and red, respectively. Non-marked values indicate results in which no statistically significant differences between MPS and HDFa were determined.

Transcript	log_2_FC > 2 or <−2 of Selected Transcripts’ Levels in Particular MPS Type vs. HDFa Line										
	I	II	IIIA	IIIB	IIIC	IIID	IVA	IVB	VI	VII	IX
Protein folding (GO:0006457)											
*CLU*	2.69	2.97	1.59	1.75	2.95	1.14	1.73	3.46	2.58	2.08	3.44
*HSPB6*	1.90	2.03	0.14	2.29	2.27	0.38	1.42	2.37	1.47	1.55	1.57
*FKBP1C*	−0.14	−0.03	−2.13	−0.27	−0.93	0.13	0.01	−0.01	−0.05	−0.55	−0.19
Response to endoplasmic reticulum stress (GO:0034976)											
*LMNA*	3.03	2.39	2.56	2.28	2.58	2.38	2.55	2.27	2.40	1.58	2.29
*CLU*	2.69	2.97	1.59	1.75	2.95	1.14	1.73	3.46	2.58	2.08	3.44
*KDELR3*	0.29	0.28	0.55	1.53	1.62	2.03	0.82	1.75	1.28	0.99	0.81
*BBC3*	1.55	1.32	1.39	1.32	1.34	2.19	1.36	1.16	1.91	1.53	2.46
*CXCL8*	−4.35	−2.46	−5.08	−0.33	−1.28	−0.12	−2.24	−1.09	−2.28	−4.04	−0.76
*CAV1*	−4.74	−2.67	−3.79	−4.75	−3.60	−1.53	−2.95	−4.74	−2.57	−1.26	−3.28
Response to unfolded proteins (GO:0006986)											
*LMNA*	3.03	2.39	2.56	2.28	2.58	2.38	2.55	2.27	2.40	1.58	2.29
*HSPB8*	0.02	0.28	−1.47	2.03	0.17	0.78	0.29	0.42	0.33	0.77	0.84
*COMP*	2.33	3.24	0.50	1.27	3.17	4.75	2.58	1.67	3.60	0.00	2.18
*HSPB7*	−0.31	1.87	0.16	1.03	0.44	2.70	1.46	0.22	1.42	1.85	1.47
*KDELR3*	0.29	0.28	0.55	1.53	1.62	2.03	0.82	1.75	1.28	0.99	0.81
*HSPB7*	−0.44	1.79	−0.14	0.96	0.37	2.38	1.01	0.07	1.43	1.72	1.81
*BBC3*	1.55	1.32	1.39	1.32	1.34	2.19	1.36	1.16	1.91	1.53	2.45
*CXCL8*	−4.35	−2.46	−5.08	−0.33	−1.28	−0.12	−2.24	−1.09	−2.28	−4.04	−0.76

## Data Availability

RNA-seq raw results are deposited in the NCBI Sequence Read Archive (SRA), under accession no. PRJNA562649mRNA. Other raw results are available from the authors upon request.

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
