# Peer review of "Decreased Levels of Chaperones in Mucopolysaccharidoses and Their Elevation as a Putative Auxiliary Therapeutic Approach"

_pharmaceutics, 2023, doi:10.3390/pharmaceutics15020704_

Round 1

Reviewer 1 Report

This work identified the reduction in heat shock protein 40 and 70 levels as a possible determinant of the dysregulated metabolism of glycosaminoglycans in mucopolysaccharidoses. This conclusion was the result of a screening study on fibroblasts obtained by patients or, in one type of rare mucopolysaccharidosis, from an animal model. The rationale of the study is well illustrated. Results are of interest. The major weakness of the study is that authors considered just one control cell line, which makes their findings more uncertain. I have no major comments apart from the previous one. Other suggestions:

-          Insert a statistical section in methods (2.8).

-          The statistical findings should be reported also in the main text, results section.

-          The statistical tests should be reported also in the figures/tables legends.

-          Scale bars for photomicrographs are missing or not very visible.

-          Fix symbols in lines 444, 448, 449 (°C).

Reviewer 2 Report

Overall, the manuscript is conceptualized well and laid out in a way that is easily readable/understood. The results presented by the authors show that in the mucopolysaccharidoses, there is a significant reduction of two heat shock proteins, but that their decreased levels were not affected by reduction in glycosaminoglycans. While these findings are valuable and interesting, the significant data arises from the transcriptomic analysis and what this data could provide to the field regarding biomarkers in therapeutic development or understanding each diseases mechanism of action in different pathways.

I believe this manuscript will benefit from a significant read-through with particular attention paid to errors in sentences, grammar, and consistency (i.e., whether all details of experiments match between the sections of the manuscript). I have identified many of these below, but not all.

Introduction: I think it would be relevant to add in some background on heat shock proteins specifically and not just chaperones, as this is a major focus of this manuscript.

If possible, it would be preferable if the non-digitized images of the western blots that show the bands in their natural state were included in the manuscript. With the recent controversy regarding western blot image manipulation, the original images should be included/presented instead. This is especially concerning when looking at the variations in gel loading, which is evident in Figure 2 when looking at the GAPDH bands.

Line 139 and Line 275 do not match regarding methods. Is the genistein concentration used 50 uM or 50 mM? Line 275 also needs to be corrected as western blot is misspelled. Further, throughout the manuscript "western blot" does not need to be capitalized or hyphenated.

Line 184 - missing period.

Line 187 - Transcriptomic analyses.

Rewrite Lines 237-240. Too many parentheses. Maybe, "Next, we investigated whether reduction of GAGs in MPS cells utilizing Aldurazyme (MPS I) Elaprase (MPS II), or genistein (an SRT-based treatment) can correct observed decreased levels of the tested Hsp proteins". And then proceed to the next sentence to say, "We found that reduction of GAG levels by the aforementioned compounds resulted in only a partial increase (how much?, give a numerical value) ...

Line 248. Remove the word "suffering" and just say "derived from patients with MPS I, II....

Line 253. Change "When analyzing" to "Further analysis of transcripts whose expression...."

Figure 4 should be labeled with (A), (B), etc. and described in the legend. Also, is there any literature regarding the variation observed in both Hsp70 and Hsp40 in the liver and brain? If these protein variations were truly correlated to the enzyme loss, then shouldn't these levels be both up- or down-regulated and not opposites?

Figure 6 may need to be redone in the final manuscript to improve the quality of the text in the actual image.

All figures: Include a brief (bolded) summary sentence at the beginning of the legend in each figure to describe what the figure is showing the reader.

Paragraph 426-434 needs to be rewritten. There are significant number of grammatical and typographic errors.

Line 461 - remove the word organism. Also remove it throughout the document when referring to the patient. Just say "patient" without the word organism.

Sentence starting in 464. Is there a source for this statement?

Sentence 477 needs to be rephrased. Try, " Using transcriptomic analysis, the present study led to the identification of genes whose products are associated with..."

Line 500 - no need for commas after proteins or Hsp40. Also at the end of the sentence, add "diseases" or "disorders" after MPS.

Reviewer 3 Report

Dear Authors, 

in this article it is shown, through an elegant analysis and a review of the literature, how the reduced levels of the main chaperone proteins, Hsp70 and Hsp40, can be a further cause of low activity or inactivity of the deficient enzymes in MPS. Furthermore, this reduction in chaperone levels could cause enzyme replacement therapy treatment to fail to lower GAG levels

The same topics have recently been covered in a recent review which I think would be useful to consider in discussion (Losada Díaz JC et al Int J Mol Sci. 2019).

The discussion should be made shorter and focused on aspects inherent in lysosomal storage diseases, without going into the pathophysiological mechanisms by which chaperones are involved in other diseases.

Minor point

Between line 426 and 433 punctuation needs to be checked
